# Population-based otoscopic and audiometric assessment of a birth cohort recruited for a pneumococcal vaccine trial 15–18 years earlier: a protocol

Kenny Chan  ,[1,2] Phyllis Carosone-Link,[3] Mary Thatcher G Bautista,[4] Diozele Sanvictores,[4] Kristin Uhler,[5,6] Veronica Tallo,[4] Marilla G Lucero,[4] Joanne De Jesus,[4] Eric A F Simoes[7]

For numbered affiliations see end of article.

**Correspondence to**
Dr Kenny Chan;
kenny.chan@childrenscolorado.org

## ABSTRACT

**Introduction** A cohort of 12 000 children in the Philippines who had enrolled in a 2000–2004 (current ages 16 to 20 years) Phase 3 11-valent pneumococcal conjugate vaccine for the prevention of radiographically confirmed pneumonia are now being asked to participate in a separate study (expected completion date September 2021) to assess the cohort's current long-term audiometric and otologic status. This new study would allow assessments of the utility of the pneumococcal vaccine in conferring its protective effects on the long-term sequelae of otitis media (OM), if any. Lack of trained local healthcare providers in otolaryngology/audiology and testing equipment in Bohol, Philippines, necessitates the development of a distinct methodology that would lead to meaningful data analysis.

**Methods and analysis** Reliable data collection and transfer are achieved by a US otolaryngologist/audiologist team training local nurses on all procedures in a didactic and hands-on process. An assortment of portable otolaryngologic and audiologic equipment suitable for field testing has been acquired, including an operating otoscope (Welch-Allyn), a video-otoscope (JedMed), a tympanometer with distortion product otoacoustic emission measurements (Path Sentiero) and a screening audiometer (HearScreen). Data will then be uploaded to a Research Electronic Data Capture database in the USA.

Tympanometric and audiologic data will be codified through separate conventional algorithms. A team of paediatric otolaryngology advanced practice providers (APPs) have been trained and validated in interpreting video otoscopy. The protocol for classification of diagnostic outcome variables based on video otoscopy and tympanometry has been developed and is being used by APPs to evaluate all otoscopy data.

**Ethics and dissemination** The study was approved by the Research Institute of Tropical Medicine, Alabang, Manila, Philippines, and the institutional review board and the Colorado Multiple Institutional Review Board of the University of Colorado School of Medicine, Aurora, Colorado, USA.

Research results will be made available to children and their caregivers with abnormal audiologic outcomes, the funders and other researchers.

## Strengths and limitations of this study

► This large-scale study involves data collection in the Philippines and data transfers to the USA; thereby, a completely novel population-based audiometric and otoscopic assessment methodology needs to be developed.

► This methodology allows for the efficacy evaluation of a randomised controlled trial of pneumococcal vaccine conducted between 2000 and 2004 on the long-term complications of otitis media (OM) (chronic suppurative OM and its sequelae on the hearing).

► Despite in-depth training of local nurses to collect data and USA's advanced practice providers to categorise video-otoscopic data, this methodology contains some degree of human factors and subjectivity that may lead to bias.

► This methodology lacks the ability to resolve inferior quality data or missing data due to logistic and geographic constraints resulting in decreased usable data for analysis.

**Trial registration number** ISRCTN 62323832; Post-results.

## INTRODUCTION

Otitis media (OM) and its most consequential complication, chronic suppurative OM (CSOM), are disproportionally overrepresented in developing countries, particularly in Asia and Sub-Saharan Africa.[1] The burden of disease is a spectrum of complications, which include hearing loss, tympanic membrane (TM) perforation, otorrhoea, adhesive OM/retraction pocket, cholesteatoma and intratemporal and intracranial complications, which in turn could have important downstream social, educational and vocational impacts. OM global health initiatives and clinical research in these populations mandate

accurate epidemiologic assessments in low-income and middle-income countries (LMICs).

However, large-scale otoscopic and audiometric assessment of ear disease in children in LMIC is difficult due to logistical constraints, including a lack of portable equipment and trained audiologists and otolaryngologists with the time and interest for these studies. A survey[2–8] of published reports in the past 15 years with relevance to OM through population studies in LMIC has revealed one or more methodologic deficiencies, namely, small sample size, lacking either otoscopic or audiometric evaluation, and particularly, assessment of CSOM.

Two recent studies have overcome these apparent deficiencies, including the documentation of frequencies of CSOM through large-scale population surveys in at-risk regions, including Asia (Indonesia)[9] and Africa (Kenya).[10] However, both of these studies relied heavily on onsite local highly trained otolaryngologists and audiologic personnel. In the Indonesian study, locally trained certified audiologists conducted hearing testing while in the Kenyan study; trained specialised medical officers were used to carry out clinical assessments and audiometric assessments. Both studies required significant time commitments of local trained audiologic personnel and senior otolaryngologists to conduct the studies. Both studies were carried out by research teams visiting schools and conducting cross-sectional studies over 3–5 days at each school that lasted for several months. However, while this strategy works for time-limited cross-sectional studies in LMIC, this strategy will not work in long-term longitudinal prospective studies.

## BACKGROUND

We had to develop a novel methodology to study the impact of an 11-valent pneumococcal conjugate vaccine on long-term audiometric and otologic outcomes of a birth cohort of 12 000, previously enrolled in a randomised clinical trial,[11] 15–18 years prior. The study is being conducted on Bohol island, Philippines, with limited otolaryngology (two general otolaryngologists on the island) and audiology (one facility with a hearing booth) resources. After piloting hands-on training of local nurses in Bohol, Philippines, to conduct otoscopic and audiologic assessments,[12] we have refined and standardised all research definitions and protocols outlined in this manuscript.

## Study Site

Bohol, Philippines, is located 640 km south of Manila and its land mass consists of Bohol island and 75 minor surrounding islands (total area of 4 821 km$^2$). It has a population of over 1.3 million inhabitants (2015 census) with a population density of 270/km$^2$. The local economy is sustained by tourism and agriculture. The 2018 poverty rate (defined as a family of five earning less than PhP 7337 a month (equivalent to US\$144)) in Bohol was reported to be 21.1% (http://www.psa.gov.ph/poverty-press-releases/nid/138411).

## Subjects of the study

The study cohort consists of a potential of over 12 000 Filipino children who were enrolled in a pneumococcal vaccine trial conducted in Bohol 2000 to 2004 between the ages of 6 weeks and 10 weeks. For the current study, these same subjects will be approximately between 16 years and 20 years of age.

## Study Aim

The specific aim of the methodology is to obtain otoscopic and audiologic data from enrolled subjects to enable assignment of OM-related and audiometric diagnostic outcomes to each of the study subjects. To achieve this goal, the methodology has the following key components: (1) portable testing equipment, (2) training of local research personal using the testing equipment, (3) development of a protocol for data collection, storage and transfer, (4) cataloguing tympanometric data using published metrics, (5) video-otoscopy data and diagnosis outcome classifications and (6) training USA's advanced practice providers (APPs) in video otoscopy. A detailed description of each of the steps is further elucidated below.

## Testing equipment

Each testing device was chosen for its portability, technical capabilities and potential reliable performance in the tropics, particularly, being able to be transported for field use.

1. An otoscope with an operating head (21700, Welch-Allyn, Skaneateles Falls, New York, USA) was selected to initially obtain a sufficient view of the TM for subsequent video otoscopy. Associated micro-instruments were selected to clear cerumen if the tympanic membrane (TM) could not be adequately visualised.
2. A portable, rechargeable battery-operated video otoscope able to capture and record high-definition (1080 p) still images and streaming video was chosen (Horus+HD Video Otoscope, JedMed, St Louis, Missouri, USA). Images are initially recorded on SD cards in the field and subsequently transferred to hard discs at the research office.
3. A portable, rechargeable battery-operated tympanometry and distortion product otoacoustic emission (DPOAE) device (Path Sentiero, model SOD07, Path Medical, Germering, Germany) was selected. While the device has many functionalities, only a limited number were required for the study. The 226 Hz frequency tympanogram issued at a pressure range of −600 daPa to +400 daPa. The tympanometer outputs for each ear include a tracing viewable by the tester as well as physical measurements of ear canal volume (ml), peak pressure (daPa), peak compliance (ml) and tympanic width (daPa). DPOAE is used as a screening tool and measured at 2 kHz, 3 kHz and 4 kHz for each

ear. Although detailed results are available through the device, only the basic results with outputs as 'pass' or 'refer' are to be used since DPOAE is only being used as a screening tool for hearing loss.

4. A portable, rechargeable battery-operated screening audiometry system (HearScreen, HearX, Pretoria, South Africa) has been chosen coupled with a smartphone (Samsung Galaxy 12) and an ISO calibrated noise-cancelling headphone (Sennheiser HD280-pro Circumural Headphone (0.5–8 kHz)). The device uses an automated testing protocol to test pure tone frequencies at 0.5 kHz, 1 kHz, 2 kHz, 3 kHz, 4 kHz, 6 kHz and 8 kHz for each ear at sound intensity levels of 20–70 dB HL. The device then generates a pass or fail interpretation based on published best practice guidelines. Subjects that fail the screening audiogram are referred for a diagnostic audiogram at Bohol Hearing Centre. All HearX data are stored on remote servers and can be accessed by the research team.

### Training of local research personnel

The training of local Filipino nurses by a US otolaryngology/audiologist team has been described previously.[12] Briefly, four nurses were trained didactically on the topic of OM and the basics of audiology and its relationship to OM clinically. Then they were given hands-on training in otoscopy, cerumen removal, video otoscopy and all of the audiologic equipment to perform tympanometry, DPOAE and screening audiograms. The training session culminated in the nursing team obtaining video-otoscopic, audiologic and clinical data from 47 subjects under direct supervision of the US otolaryngologist/audiologist team.

### Protocols for data collection, storage and transfer protocols

Research data collection by the local research team include: (1) case report forms; (2) video-otoscopy and audiologic testing and 3) miscellaneous data.

### Case report form (CRF)

Each research subject has an original paper CRF, which contains patient identifiers and clinical information entered by the research personnel following a parental interview with ear-specific medical history (ie, ear infection, discharge, TM perforation and hearing loss). CRFs collected during the week are uploaded by the research nurse into a Research Electronic Data Capture (REDCap) database hosted at the University of Colorado Denver School of Medicine, Aurora, Colorado, USA.

### Video-otoscopy and audiologic testing

Subjects whose TMs are devoid of cerumen or whose cerumen can be cleaned by the research nurses undergo video-otoscopy and audiologic testing. All subjects whose cerumen cannot be cleaned adequately are given a bottle of mineral oil and a medicine dropper and advised to use 5 drops twice a day for 7 days and asked to return for another assessment. However, subjects that fail to have adequate cerumen removal after the subsequent visit or

had active otorrhoea are referred to the local otolaryngologist for further management

The research nurses record the two ears of a research subject in a regimented sequence for ease of review by the APPs at a subsequent time (right ear still photo, left ear still photo, right ear video and left ear video). SD cards are downloaded at the end of each weekday and images stored on two hard disks (original and backup disks). Data are transferred to the USA by external drives.

Audiologic testing data are processed in the following manner. The four discreet outputs produced by the tympanometer are used to determine tympanogram types. An overlay 'rectangle' onto the tympanometer tracing is used by the manufacturer to include most 'normal' tympanograms within these confines as defined by Jerger[13] to assist with diagnosis. The local research team uses this real-time judgement call for a 'pass' or 'fail' depending if the peak of the tracing is within the 'normal' ranges as dictated by the 'rectangle'. DPOAE results are also automatically determined by the device as either 'pass' or 'refer'. Hearing screening done on subjects using the HearScreen device also classifies subjects as 'pass' or 'fail' by the device. All subjects who fail the tympanogram, DPOAE or HearScreen are referred to Bohol Hearing Centre for diagnostic audiometry. All raw tympanogram, automated DPOAE, automated HearScreen and raw HearX data are entered by the research supervisor to the REDCap database on a weekly basis.

### Miscellaneous data

All audiologic data from Bohol Hearing Centre, as well as all otologic diagnoses made by the island otolaryngologist, are also entered into the REDCap database by the research supervisor on an as-needed basis.

### Cataloguing of tympanometric data

The tympanometric data for each ear consisting of ear canal volume (ml), peak pressure (daPa), peak compliance (ml) and tympanic width (daPa) are sorted based on the classic classification established by Jerger: A, B, C, $A_s$ and $A_d$ types. As such, a portion of data would result in tympanometric parameters landing outside of the confines for each type defined by Jerger.[13] Since the research team viewed it imperative that each video otoscopy be accompanied by a tympanogram type to aid in video-otoscopy interpretation, an ad hoc methodology had to be developed to capture the 'non-classifiable' tympanograms. The study otolaryngologist and a senior audiologist independently will read all 'non-classifiable' tympanograms, and discordant classifications will be adjudicated by a second senior audiologist. The majority vote will confirm the classification type.

### Video-otoscopy data and diagnosis outcome classifications

Video-otoscopy analysis leading to disease classification takes into account all otoscopic abnormalities associated with acute OM, chronic non-suppurative OM and CSOM, as well as a non-OM diagnoses (eg, otitis externa)

**Box 1** List of video-otoscopic finds depicting physical findings associated with otitis media

- ► Redness of tympanic membrane (TM).
- ► Opacification not due to scarring (dullness, thickened and webbing).
- ► Bubbles or air-fluid interfaces or fluid.
- ► Abnormal TM colour: white, yellow, amber or blue.
- ► Fullness or bulging of TM.
- ► Thinned out area of TM or healed perforation seen in 360°.
- ► Tympanosclerosis/scarring.
- ► Perforation small (<25% of TM).
- ► Perforation large (≥25% of TM).
- ► Retraction/retraction pocket/generally thinned out but not healed TM.
- ► Retraction/pocket bottom not seen.
- ► Otorrhoea not due to otitis externa (moisture).
- ► Granulation tissue or squamous debris.
- ► Otitis externa.

summarised in box 1. APPs document findings rather than immediately arrive at a diagnosis. Each video-otoscopic image is associated with a tympanogram type as to simulate pneumatic otoscopy, a proxy for the gold standard to diagnose middle-ear effusion.[14 15] The gold standard for the presence of middle-ear fluid is detected only by myringotomy; however, this being impossible, the proxy methodology used here is based on the decisions made by the senior paediatric otolaryngologist with over 30 years of performing otologic procedures for complications from OM, including tympanoplasties and mastoidectomies. Given it would be impractical and impossible for the paediatric otolaryngologist to view thousands of video otoscopies, experienced otoscopists (APPs) have been trained and validated in identifying otoscopic abnormalities listed in box 1.

All video otoscopies that are deemed as poor quality by the APP are reviewed by the otolaryngologist of the study and would remained classified as poor quality or into one or more of the following TM findings.

Video-otoscopic findings for each ear will then merged with the four tympanometric output parameters and clinical medical history to arrive at discreet diagnoses (figure 1). Four broad diagnostic categories with definitions include normal, mild, moderate and severe ear disease. If two ears of a subject have disparate levels of middle-ear disease, the subject would be classified as having the more serious form of ear disease for the purpose of analysis.

## Normal

No evidence of ear disease as determined by no disease-related video-otoscopic findings regardless of tympanometric abnormal states and clinical medical history of ear disease.

## Mild Ear Disease

- ► Acute OM: the presence of at least two of the following three otoscopic signs: (1) white, yellow, amber or blue

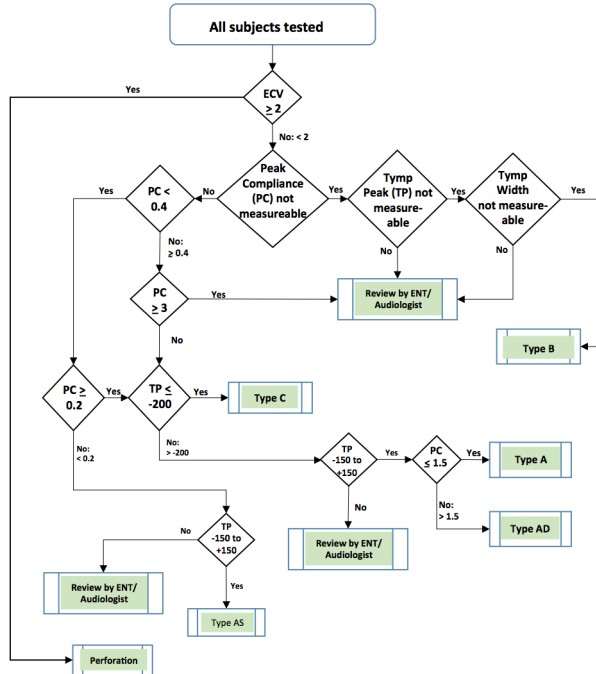

**Figure 1** Schema in classifying tympanometric types based on tympanometer outputs (accordance with Jerger classification). ECV, external canal volume; ENT, Ear, Nose, and Throat physician, or Otolaryngologist.

TM (ie, abnormal colour), (2) opacification not due to scarring, (3) tympanometry peak compliance <0.2 mL or tympanometric width >200 daPa on tympanometry, or bubbles or air-fluid interfaces, and one of the following three signs/symptoms: (a) ear pain, (b) redness of the TM and (c) fullness or bulging of the TM or acute purulent otorrhoea not due to otitis externa.[16–18]
- ► OM with effusion (OME): the presence of at least two of the following three otoscopic signs: (1) white, yellow, amber or blue TM (ie, abnormal colour), (2) opacification not due to scarring, (3) tympanometry peak compliance <0.2 mL or tympanometric width >200 daPa from tympanogram, or bubbles or air-fluid interfaces, and the absence of the following four signs/symptoms: (a) ear pain, (b) redness of the TM, (c) fullness or bulging of the TM or (d) acute purulent otorrhoea not due to otitis externa.[16–18]
- ► Healed perforation: otherwise normal TM with a thinned-out area of TM.
- ► Myringosclerosis: a hardening of the ear drum as calcium deposits form on the ear drum and middle ear, peak compliance on tympanogram >5.0 mL.

## Moderate ear disease

- ► Dry perforation: perforation of the TM without otorrhoea or ear canal volume ≥2 mL on tympanogram.
- ► Adhesive OM: intact TM but retraction or retraction pocket seen.

## Severe ear disease

- ► CSOM, active: perforation of the TM with otorrhoea duration of more than 2 weeks, or TM with retraction

pocket unable to see deepest part, with granulation tissue, squamous debris or otorrhoea.

## Protocol to deal with ears with incomplete data set

It is expected some subjects or ears would not have a complete data set. Reasons would include poor quality collection techniques, uploading error and the subjects not following through with seeing the island otolaryngologist or completing the audiogram at Bohol Hearing Centre.

When the CRF indicates otorrhoea and no other data are available/discernable, these ears are classified as CSOM. When the case report form indicates no otorrhoea or ear pain and only the tympanogram is available, the ears are classified either as normal or OME based on type A or types B/C, respectively. In ears without tympanograms, ear disease classification is solely based on video otoscopy. When the CRF notes no ear pain, tympanogram is missing and video otoscopy is missing/undiscernible, these ears are deemed 'missing data' and will not be included as part of the final analysis.

## Training of USA's APPs

Three outpatient APPs from the Department of Otolaryngology at Children's Hospital Colorado with varying length of tenure and training in otoscopy (≥7 years of clinical experience) were trained in reading video-otoscopic images using didactic in-person training methods to identify the 14 possible otoscopic signs described in box 1.

## Patient and public involvement

No patient and public involvement took place in the design, or conduct, or reporting, or dissemination plans of the research.

## Statistical analysis methodology

Demographic attributes and potential risk factors derived from the Parent Questionnaire will be analysed to determine which ones should be included in the multivariate analyses. A p value <0.15 will be used to select the 'significant' variables. ORs will be computed for the dichotomous categorical variables, using a Cochran's Mantel-Haenszel statistic to compute the p value. Continuous variables and categorical variables with more than two categories will be analysed using Student's t-test or Wilcoxon rank-sum test. A rank-sum test will be used when the assumptions required for the t-test are not met. When there are significant differences in ear disease classification between the vaccine and placebo group, then a multivariate logistic regression model will be derived to identify the vaccine efficacy for the 11-valent pneumococcal conjugate (11PCV) vaccine, and significant effects for the covariables.

## DISCUSSION

The methodology discussed in this manuscript will be sed to analyse a potential cohort of 12 000 children previously enrolled in a pneumococcal vaccine trial in Bohol, Philippines, and will now be assessed 12–18 years later for the vaccine's effects on otoscopic and audiologic endpoints. It is unique in several aspects: size of the cohort, clinical endpoints and complexity of data analysis.

The sample sizes of previously published randomised controlled pneumococcal conjugate vaccine trials have been clearly summarised by a recently published Cochrane Review.[19] Our cohort of over 12 000 children to be assessed likely would be lowered since attrition might occur due to inability to locate the families due to relocation or immigration after 15–18 years, as well as parental refusal to participate. Regardless, the biggest pneumococcal vaccine trials were published by Black et al in 2000[20–22] and Tregnaghi et al[23] involving 37 868 and 24 000 subjects, respectively. Black's study was conducted in the USA, and acute OM was the only ear-related outcome.[21] The Tregnaghi et al's study was done in Central and South American countries, but only 7214 children from Panama were included in the OM portion of the study.[23] Prymula et al published an acute OM trial using an 11-valent pneumococcal vaccine conjugated to Hemophilus influenzae-derived protein D trial in 4968 subjects.[24] This trial was conducted in the Czech Republic and Slovakia, and tympanocentesis and pneumococcal subtyping were done. A recent open label trial of maternal immunisation with a 23-valent polysaccharide pneumococcal vaccine in 227 indigenous mothers in the Northern Territory, Australia, showed no impact on OM-related infant outcomes at 7 months of age.[25]

The clinical outcomes for the randomised controlled trials have included the reduction of pneumococcal OM, recurrent OM, OM visits and tympanostomy tube insertions.[20–24 26] None of these randomised controlled trials were focused on the long-term sequalae of OM in terms of changes in the TM as evaluated by otoscopy and changes in hearing as evaluated by audiometry. While prevention of disease is compelling for the use of the pneumococcal vaccine, preventing of long-term sequalae is just as important. Trials conducted in the indigenous populations in Australia that examined long-term sequelae occurred after licensure of pneumococcal vaccines in Australia and so followed before/after designs.[27–30] Thus, our study will be among the largest randomised controlled pneumococcal vaccine trials for OM and the only one to study the long-term effects on OM sequelae 15–18 years after the initial trial.

The complexity of the methodology dealing with data lies at several levels. First, it requires the raw data collected by the local research team be uniform and robust. Despite a local research supervisor ensuring good quality data were recorded and uploaded, a certain degree of data scrubbing will be required on the US side. Tympanometric and audiologic data are codified based on published definitions of classifications. An algorithm has also been developed to derive OM diagnostic endpoints from all video-otoscopic images that was reliant on the raw images and tympanogram types. The overall construct of the methodology may serve as a platform

for conducting large scale OM research in LMIC, particularly, if the objectives include garnering otoscopic and audiologic determinates and assessment of chronic and long-term OM complications.

## ETHICS AND DISSEMINATION
### Ethics
The 'Do No Harm' precept has been inculcated in the study. Despite the healthcare discrepancy that exists between the USA and Bohol, Philippines, accommodations have been made in the methodology to refer the research subjects to local healthcare professionals. The local field workers will be instructed to refer subjects to Bohol Hearing Centre for failed OAEs, tympanometry and/or screening audiometry and to the local otolaryngologist for ear pathology (inability to remove cerumen after instillation of mineral oil from a previous visit, otorrhoea and suspicion for cholesteatoma).

The study was approved by the Research Institute of Tropical Medicine, Alabang, Manila, Philippines, and the institutional review board (20 February 2016) and the Colorado Multiple Institutional Review Board of the University of Colorado School of Medicine, Aurora, Colorado, USA (16-0473).

Study data are collected and managed using REDCap electronic data capture tools hosted at University of Colorado. REDCap is a secure, web-based application designed to support data capture for research studies, providing: (1) an intuitive interface for validated data entry; (2) audit trails for tracking data manipulation and export procedures; (3) automated export procedures for seamless data downloads to common statistical packages and (4) procedures for importing data from external sources. Data are stored, managed and processed within the system. Data are exported as SAS (version 9.4) files to secure servers on the University of Colorado and Children's Hospital Colorado systems, where data will be analysed.

### Dissemination
Research data are privately held. Research results will be made available to children and their caregivers with abnormal audiologic outcomes, the funders and other researchers. After publication, anonymised and curated data will be made available to researchers following guidelines of the Bill and Melinda Gates foundation. Data will be published in open access journals only.

**Author affiliations**
¹Department of Otolaryngology–Head and Neck Surgery, University of Colorado School of Medicine, Aurora, Colorado, USA
²Department of Pediatric Otolaryngology, Children's Hospital Colorado, Aurora, Colorado, USA
³Department of Pediatrics, University of Colorado School of Medicine, Aurora, Colorado, USA
⁴Department of Clinical Trials, Epidemiology, and Biostatistics, Research Institute for Tropical Medicine, Muntinlupa, Philippines
⁵Department of Audiology, Speech-Pathology, and Learning, Children's Hospital Colorado, Aurora, Colorado, USA
⁶Department of Physical Medicine and Rehabilitation, University of Colorado School of Medicine, Aurora, Colorado, USA
⁷Department of Epidemiology, Colorado School of Public Health, Aurora, Colorado, USA

**Acknowledgements** The authors, on behalf of study team, would like to thank the other nurses who were the subjects of training: Nino Carlo Salutan, Leslie Salutan, Marelie Dagupan and Shobelle Anunciado; the staff of Research Institute of Tropical Medicine, Manila, Philippines, for their dedication to their patients, including our trial participants; and Deborah Hayes, PhD, who was instrumental in developing the audiology protocols.

**Contributors** Conception and acquisition of funding: EAFS. Epidemiologic and field design: EAFS, VT, MGL, JDJ, MTGB and DS. Otolaryngologic study design: KC, KU, EAFS, PC-L. Audiologic study design: KU, KC, EAFS and PC-L. Analysis: EAFS, KC, PC-L and KU. Research Electronic Data Capture database creation: PC-L. Drafting and revision: all authors.

**Funding** This study was supported by the Bill and Melinda Gates Foundation (grant number: OPP1142570) and Colorado CTSA (grant numbers: UL1TR002535, KL2TR002534, TL1TR002533 and UL1 RR025780). The contents of this report are solely the responsibility of the authors and do not necessarily represent the official views of their institutions or organisations or of the sponsors. The funders did not participate in any aspect of the study, including study conduct, data collection, analyses of the data or the writeup of the manuscript. The corresponding authors confirm that they had full access to all the data in the study and had final responsibility for the decision to submit for publication.

**Competing interests** None declared.

**Patient and public involvement** Patients and/or the public were not involved in the design, or conduct, or reporting, or dissemination plans of this research.

**Patient consent for publication** Not required.

**Provenance and peer review** Not commissioned; externally peer reviewed.

**ORCID iD**
Kenny Chan http://orcid.org/0000-0001-9989-1322

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
