## [Reviewer comments · BMJ Open]

ARTICLE DETAILS

TITLE (PROVISIONAL)	Population based otoscopic and audiometric assessment of a birth cohort recruited for a pneumococcal vaccine trial 15-18 years earlier: Protocol
AUTHORS	Chan, Kenny; Carosone-Link, Phyllis; Bautista, Mary; Sanvictores, Diozele; Uhler, Kristin; Tallo, Veronica; Lucero, Marilla G.; De Jesus, Joanne; Simoes, Eric

VERSION 1 – REVIEW

REVIEWER	Harvey Coates AO DM FRACS University of Western Australia, Perth, Western Australia
REVIEW RETURNED	15-Sep-2020

GENERAL COMMENTS	This prospective study, funded by the Bill and Melinda Gates Foundation, is well designed, robust and feasible. The information obtained will be valuable and innovative.
---

REVIEWER	Marchisio Paola University of Milan , Milan, Italy
REVIEW RETURNED	29-Dec-2020

GENERAL COMMENTS	The protocol is innovative and might, indeed, add new data to the literature involving pneumococcal vaccines and otitis media. The design is accurate and well described. Training of local nurses is a great idea. The attention to the presence of cerumen (often neglected) is highly appreciable. There are a few comments which need answers ABSTRACT • Page 5, line 9: the type of pneumococcal conjugate vaccine (PCV) should be specified (as reported on page 9, line 22)• Page 5, line 9: a vaccine is usually intended to “prevention” and not “treatment”• Page 5, line 16: several studies have already assessed the efficacy of PCV on otitis media.- Therefore, the focus should be put on the long-term effects or, alternatively, on the site of the study (not industrialized country) or both. In fact, this is correctly stated on page 7, line 17• Page 5, line 37: otoscope (singular) BACKGROUND • Page 9, line 24: cognitive: please add also to abstract, or delete is not appropriate to the study METHODS AND ANALYSIS
--

	 • Page 10, line 9: is an age limit defined? Usually children are aged up to 18 years. If a child was vaccinated in 2000, then in 2020 (or 2021) he/she would be 20 years old • Page 12, line 47: otorrhea is preferable to “discharge” • Page 13. How will the recent health of included children be assessed? The protocol should include some indications about assessing the absence of infectious problems (beyond middle ear) in the last 7 – 14 days. Otherwise how can the actual assessment be related to the PCV administration? • Page 16, lines 30 and 48: acute purulent otorrhea is a sign of acute otitis media. It cannot be considered also a sign of otitis media with effusion, which, by definition, is not acute. • Statistical analysis is lacking DISCUSSION  • Page 19, line 5: Tregmaghi should be “Tregnaghi” (in the references if correct) TABLE 1  • Page 26, line 30 : is “or” correct? Should it be “of”? • Page 26, line 30: should be the margins of the perforation described? A possible indicator of the age of the perforation.
--	--

VERSION 1 – AUTHOR RESPONSE

RESPONSE TO REVIEWERS' COMMENTS

ABSTRACT

• Page 5, line 9: the type of pneumococcal conjugate vaccine (PCV) should be specified (as reported on page 9, line 22)

Added

• Page 5, line 9: a vaccine is usually intended to “prevention” and not “treatment”

Changed

• Page 5, line 16: several studies have already assessed the efficacy of PCV on otitis media.- Therefore, the focus should be put on the long-term effects or, alternatively, on the site of the study (not industrialized country) or both. In fact, this is correctly stated on page 7, line 17

Revised

• Page 5, line 37: otoscope (singular)

Changed

BACKGROUND

• Page 9, line 24: cognitive: please add also to abstract, or delete is not appropriate to the study

Deleted

METHODS AND ANALYSIS

• Page 10, line 9: is an age limit defined? Usually children are aged up to 18 years. If a child was vaccinated in 2000, then in 2020 (or 2021) he/she would be 20 years old

The ages were defined during the previous PCV vaccine trial but not defined for this study since this is a long-term follow-up study. The sentence has been amended.

• Page 12, line 47: otorrhea is preferable to “discharge”

The case study form is being used by the research nurses for patient information collection only and does not denote clinical meaning. The word “discharge” has not been changed.

- Page 13. How will the recent health of included children be assessed? The protocol should include some indications about assessing the absence of infectious problems (beyond middle ear) in the last 7 – 14 days. Otherwise how can the actual assessment be related to the PCV administration?

The PCV vaccine was administered 16 to 20 years ago, and we are not examining children longitudinally to see if there is an impact on current acute otitis media. The aim of this trial is to study the long-term effects of pneumococcal vaccine on ear disease and hearing outcomes. We do ask questions about a history of current ear pain (to differentiate between otitis externa, AOM and OME) and duration of otorrhea (acute versus chronic). Other than these symptoms we did not inquire about other recent health conditions.

- Page 16, lines 30 and 48: acute purulent otorrhea is a sign of acute otitis media. It cannot be considered also a sign of otitis media with effusion, which, by definition, is not acute.

We agree with the reviewer; acute otorrhea has been grouped under acute otitis media definition.

- Statistical analysis is lacking

We have added a section on statistical analysis of the data.

DISCUSSION

- Page 19, line 5: Tregmaghi should be “Tregnaghi” (in the references if correct)

Corrected

TABLE 1

- Page 26, line 30 : is “or” correct? Should it be “of”?

Corrected

- Page 26, line 30: should be the margins of the perforation described? A possible indicator of the age of the perforation.

While we agree with the reviewer, the investigators wanted to schematize a classification that is easy to use by the advanced practice providers and at the same time to denote the severity of the perforation and its potential effect on hearing. Therefore, we chose size of the perforation instead.

EXPLANATION OF REVISING FIGURE 1

Since submission of the manuscript, the authors simplified Figure 1. The revision places external canal volume (ECV) of > 2 ml as the first step. This does not alter the flowchart’s diagnostic definitions. It reduces the number steps and improves its readability.

Thank you again for your time and effort.